# Building a Teleost Fish Traceability Program Based on Genetic Data from Pacific Panama Fish Markets

**DOI:** 10.3390/ani13142272

**Published:** 2023-07-12

**Authors:** Edgardo Díaz-Ferguson, Magaly Chial, Maribel Gonzalez, Edgardo Muñoz, Olga Chen, Ovidio Durán, Angel Javier Vega, Carlos Ramos Delgado

**Affiliations:** 1Coiba Scientific Station (COIBA AIP), Gustavo Lara Street, Bld. 145B, City of Knowledge, Clayton, Panama City 0843-01853, Panama; 2Faculty of Natural and Exact Sciences, Department of Genetics and Molecular Biology, University of Panama, Panama City 0824-3366, Panama; 3Center of Marine Science and Limnology, Department of Marine Biology, University of Panama, Panama City 0824-01853, Panama; 4School of Biology, Regional Center, University of Panama, Santiago de Veraguas 0923-00125, Panama

**Keywords:** COI, *teleostei*, nucleotide diversity, haplotypic diversity, qPCR, demographic history, environmental DNA

## Abstract

**Simple Summary:**

Molecular identification of fish tissue samples from 203 individuals was conducted based on cytochrome oxidase I gene segment sequencing. A total of 34 species from 14 families (Ariidae, Caranjidae, Centropomidae, Gerreidae, Haemulidae, Lobotidae, Lutjanidae, Malacanthidae, Mugilidae, Serranidae, Scianidae, Scombridae, Sphyraenidae, Stromateidae) were identified at the species level from 164 obtained sequences. Three Caribbean species were also molecularly identified among the analyzed samples (*Mycteroperca xenarcha*, *Paralonchurus brasilensis* and *Lobotes surinamensis*). Species diversity was slightly higher in the Gulf of Panama than in the Gulf of Chiriquí. Genetic diversity and connectivity between Gulf areas was compared using values of haplotypic diversity and nucleotide diversity for genetic diversity and genetic distances and genetic differentiation (Fst) for connectivity. A high level of connectivity was observed between the Gulf of Chiriqui and the Gulf of Montijo, showing the existence of a single stock in that area for the following species: *Scomberomorus sierra*, *Caranx caninus* and *Lutjanus guttatus*. The demographic history of the most common species was examined, and population expansion was evidenced for two snapper species, *L. peru* and *L. argentiventris* (significant and negative values of Tajimas D). Another important contribution from this research was the design of primers and dual-labeled probes for environmental DNA and qPCR detection for three of the most abundant species.

**Abstract:**

Fish tissue samples from 203 adult individuals were collected in the main ports and markets of the Pacific coast of Panama. Molecular identification based on a cytochrome oxidase I gene segment of all species was verified by GENBANK reference sequences. A total of 34 species from 14 families (Ariidae, Caranjidae, Centropomidae, Gerreidae, Haemulidae, Lobotidae, Lutjanidae, Malacanthidae, Mugilidae, Scianidae, Scombridae, Serranidae, Sphyraenidae, Stromateidae) were identified at the species level from 164 sequences. Additionally, three Caribbean species were molecularly identified among the analyzed samples (*Mycteroperca xenarcha*, *Paralonchurus brasilensis* and *Lobotes surinamensis*). Species diversity was slightly higher in the Gulf of Panama than in the Gulf of Chiriquí. For species with five or more individual sequences, genetic diversity and genetic connectivity parameters such as total number of haplotypes (*H*), haplotype diversity (*Hd*), and nucleotide diversity (π) were calculated. Overall, pelagic-migratory species showed higher values of genetic diversity than coastal and estuarine species with some exceptions. Connectivity between Gulf areas was compared using values of genetic distances and genetic differentiation (*Fst*). The high level of connectivity observed between the Gulf of Chiriqui and the Gulf of Montijo indicates the existence of a single stock in that area for the following species: *Scomberomorus sierra*, *Caranx caninus* and *Lutjanus guttatus*. The demographic history of the most common species was examined using Tajima’s D values, suggesting population expansion for two snapper species, *L. peru* and *L. argentiventris*, having significant and higher values. Another important contribution from this research was the production of primers and dual-labeled probes for environmental DNA detection using qPCR for the five most abundant species (spotted rose snapper, yellow snapper, green jack, Pacific crevalle jack and the Pacific sierra fish). These markers represent a new set of tools for environmental DNA (eDNA) detection and molecular traceability of three commercially important fish species along the supply chain including landing sites and markets of the main fishery areas.

## 1. Introduction

The advent of the genetic barcode has improved the capacity to resolve the miscalling of species; detect seafood and product fraud; minimize trafficking of endangered and CITES (International Convention for Endangered Species Trade) species; and reduce illegal, unregulated and unreported fishing (IUU) [1,2]. In addition, a barcode has allowed the identification and discovery of new species, separation of cryptic species, species complexes, stock characterization, detection of molecular hybrids and understanding migrations and connectivity patterns [3,4]. In fishes, the use of a barcode began in 2003 [5]; and by 2009, more than 5000 fish species were characterized by this method [6]. Considering that Osteichthyes (boney fishes) are the most diverse vertebrate taxa, with more than 28,000 species reported [7,8], it is evident that scientists still have some important work to carry out.

Panamanian fish fauna is highly diverse and includes 1412 species, of which 1240 are marine species [9,10]. Approximately, 814 of these species have been reported in the Pacific coast of Panama and this represents 60% of the fish diversity for the country, making it the most diverse vertebrate taxa [7]. In a recent review, a total of 223 commercial fish species were reported for the Pacific of Panama, corresponding to 183 bony and 70 cartilaginous fish [10]. These species are locally commercialized or exported as fish products (fillets, fins, and whole fish). However, the final destiny of some of the most commercially important fish species (i.e., snappers, drummers, pacific sierra, groupers, queen corvina, dolphin fish, tunas, jacks and sharks) and their derived products is currently unknown or partially understood due to taxonomic identification mistakes and the lack of species-specific tariff codes that should be based on a binomial nomenclature species (BNS) system that is the key for the establishment of species-specific traceability program. In some cases, lack of this BNS is usually magnified by clustering three or more species under the same tariff code. For example, Pacific Panama has 10 species of lutjanids that are clustered under snappers as unique tariff code whereas 37 species of shark and 22 species of rays are clustered under the designation “shark” as tariff code [11,12]. These taxonomic identification mistakes lead to an improbable determination of which species are certainly exploited, overexploited, and exported. Even in food products such as “ceviche” (a mix of raw fish cooked with lemon) which is popular among local consumers; contains species that are currently unlisted, invasive, CITES or under any of the 12 IUCN categories such as sharks, rays and billfishes [13].

Traceability is a records-based system defined as the ability to trace the history, origin location and application of a traceable resource unit (TRU) which is a species or fishery product along the food supply chain using physical (electronic tags), chemical (isotopes) or molecular methods (i.e., DNA barcode or DNA fingerprint) [14,15]. In the 21st century, food safety, transparency and product quality control are a competitive advantage in a globalized market that relies on traceability [16,17]. In addition, transparency in fisheries is considered one of the main challenges that is currently facing the fishery industry [18].

The top 21 countries for food production and consumption participating in the Organization for Economic Co-Operation and Development (OECD) were ranked based on the nature or scope of their mandatory traceability regulations. Results showed that mandatory legislation for food and feed is only established in EU and Pan EU countries. Australia, New Zealand, Canada, Japan, Brazil and the United States have an overall world score ranking of “average”. For Russia insufficient data were received and China was ranked as poor. Interestingly, Costa Rica was recently included in the OECD without having a national traceability system for fish and fish products. However, no information or the species included in this traceability system is available.

The implementation of a traceability system in Latin America for fish and fishery products is currently in an early development phase or yet to be implemented [19]. Only countries such as Peru, Chile and Ecuador have implemented a traceability model for some fishery products [20]. Nonetheless, these traceability programs are partially applied to some species or taxa, or they are only utilized for importation but not for domestic commercialization [21]. When traceability is applied, only physical traceability methods (labeling the products) are employed, and no chemical or molecular traceability methods are currently in use for the public sector [22,23,24]. In fact, for the most common commercial species listed in Central America and the Eastern Pacific from Mexico to Ecuador, stock composition and traditional population dynamics studies have been uniquely conducted in a handful of species, e.g., red snapper, *Lutjanus peru* [25]; spotted rose snapper, *Lutjanus guttatus* [26]; green jack, *Caranx caballus* [27]; Pacific sierra, *Scomberomorus sierra* [28]; dolphin fish, *Coryphaena* hippurus [29]; Wahoo, *Acanthocybium solandri*; and the yellowfin Tuna, *Thunnus albacares* [30]. Likewise, genetic data for the species listed just before are only available for the Mexican populations of spotted rose snapper and red snapper [31,32], Pacific sierra [33] Wahoo [34] and the main tuna species [30]. On the other hand, molecular identification methods have been employed for traceability along the commercial chain and for mislabeling purposes in Caribbean populations of *Lutjanus campechanus* and *Lutjanus purpureus* [35,36].

The main objectives of this research are to provide a DNA barcode list of Pacific Panama commercial fish and the first haplotype candidates for molecular traceability as well as the first values of genetic diversity, connectivity and demographic history for the most common species collected in the largest fishery areas of Pacific Panama (Gulf of Panama and Gulf of Chiriqui). This is also a pioneering study, providing potential markers for the most common species.

## 2. Materials and Methods

### 2.1. Sample Collection on Local Markets and Fish Gathering Centers

Main fishery landing ports of the Pacific Panama (Coquira, Pedregal and Mutis) and the biggest fish gathering centers of the Pacific side of the Country (Mercado del Marisco-San Felipe Market, Mercado de Anton, Mercado de Santiago and Isla Verde-Golfo de Montijo) were visited (Figure 1). Additionally, red snapper samples from Pixvae area, located within the buffer zone of Coiba Island National Park (largest marine park in Central America and UNESCO Natural Heritage Site), were also analyzed. At each collection site, fin clips and muscle tissue of the main fish families reported as commercial for the Pacific Panama were collected and stored in vials containing ethanol at 95%. Each sample was labeled and recorded using the species common name, possible fishery zone (Gulf) and the landing site name.

### 2.2. DNA Extraction and PCR Amplification

A total of 203 fish tissue samples were collected. DNA was extracted from all these samples by different methods including the use of a DNA easy kit for blood and tissue (QIAGEN, Inc., Valencia, CA, USA) as well as the phenol-chlorophorm method. DNA was quantified using a nanodrop (Thermo Scientific, Waltham, MA, USA, https://www.thermofisher.com/pa/en/home.html, accessed on 2 July 2023). PCR reactions were conducted with 1 µL of each primer using fish universal primers C_FishF1 and C_FishR1 [37], 8.5 µL of molecular-grade water, 2 µL of template DNA and 12.5 µL of 2× master mix reagent (QIAGEN, Inc., Valencia, CA, USA). All PCR reactions were conducted for 35 cycles at an annealing temperature (Ta) of 55 °C [37].

### 2.3. Sequence Analysis

PCR products were visualized in 2% agarose gels. Positive products were cleaned and sequenced by Macrogen (Rockville, MD, USA). Sequences were verified for size and checked for quality using Geneious Prime [38]. High-quality sequences were trimmed, edited to a size of 614 bp and aligned for final analysis in Geneious Prime and DnaSP version 6.1. Edited sequences were imported to other programs for further genetic diversity and connectivity analysis as Fasta files. GenBank accession numbers for all obtained sequences were generated through Sequin software version 15.50. Sequence identity was verified to the species level using the BOLD taxonomic and reference sequence search [39].

### 2.4. Species and Genetic Diversity in Panama Pacific Fisheries

#### 2.4.1. Species and Family Diversity

The total number of families and species richness (*S* = total number of species per collection site) were quantified overall (all samples considered) and by site and by gulf (Gulf of Panama, Gulf of Chiriqui and Gulf of Montijo) (Table 1). Total number of species and families by gulf was compared using a non-parametric test (Kruskal–Wallis).

#### 2.4.2. Genetic Diversity

Genetic diversity of the most common species was determined overall and by site. Genetic diversity was estimated using the following parameters: total number of haplotypes (*H*), total number of polymorphic sites (*S*), haplotype diversity (*Hd*) and nucleotide diversity (*π*). These values were determined in species with more than five sample sequences available by sampling site using DnaSP v. 6.1 [40].

#### 2.4.3. Genetic Connectivity and Demographic History

Genetic differentiation (*Fst*) was calculated overall by species and by site to examine the existence of genetic stocks and understand patterns of connectivity and differentiation among sites (Table 2 and Table 3). Hudson 2000 statistical (snn) significance *p* < 0.0001 was also compared among species to infer genetic structure and connectivity. The total number of haplotypes (*H*) and haplotype diversity (*Hd*) were used as a measure of population identification, stock fitness and sustainability. Connectivity between sites was estimated indirectly using values of pairwise *Fst* and genetic distances (species with representative sequences in all sites) were calculated using DNAsp 6.0.

Demographic history was estimated using values of Tajimas D and Fu’s. These parameters were calculated in species with more than 10 individual sequences. Population expansion and bigger population size was evidenced with negative and significant values of Tajima’s D while positive and non-significant values of this parameter showed smaller population size and population contraction.

#### 2.4.4. Developing Tools for Molecular Taxonomy and Traceability

Primers and dual-labeled probes for TaqMan qPCR environmental DNA detection and quantification were developed in silico using two programs Genescript and Primer Express 3.0. Ten sequences from each species (*Caranx caballus*, *Caranx caninus*, *Lutjanus guttatus*, *Lutjanus argentiventris*, and *Scomberomorus sierra*) including sequences from this study and GenBank reference sequences were aligned and compared to find a conserved specific region for eDNA target species detection. Amplicon segments were tested through BLAST to corroborate species specificity as well as primer and probe set.

## 3. Results

### 3.1. Sequence Analysis

We obtained 164 partial sequences of 614 base pairs of the mitochondrial gene cytochrome oxidase I out of 203 tissue samples. For each identified haplotype of the analyzed species, a GenBank accession number was generated for all sequences (Table 1). The analyzed sequences had a query call between 98 and 100% and 100% similarity with GenBank reference sequences. Some sequences reported here represent the first sequences of the species for Panamanian waters (Table 1).

### 3.2. Species, Family and Genetic Diversity in Panama Pacific Fisheries

A total of 34 species belonging to 14 families were identified out of 164 sequences obtained from 223 fish tissue samples. The most diverse families were Carangidae (*Selene peruviana, Caranx caballus*, *Caranx caninus*, *Carangoides vinctus* and *Caranx sexfasciatus*) and Scianidae (*Nebris occidentalis*, *Larimus pacificus*, *Cynoscion phoxocephalus*, *Cynoscion leiarchus* and *Paralunchurus brasiliensis*), with five species each. The second most diverse family was Lutjanidae (*Lutjanus argentiventris*, *Lutjanuns guttatus*, *Lutjanus peru* and *Lutjanus colorado*); and Centropomidae (*Centropomus robalito*, and *Centropomus viridis* and *Centropomus unionensis* and *Centropomus medius*), with four species each. Haemulidae family had three species (*Pomadasys panamensis*, *Haemulopsis leuciscus* and *Orthoprisitis chalceus*). Other observed families in this study such as Lobotidae (*Lobotes surinamensis* and *Lobotes pacificus*), Scombridae (*Scomberomorus sierra* and *Thunnus albacares*) and Serranidae (*Myctoperca xenarcha* and *Hyporthodus acanthistius*) had two species each. Finally, families: Sphyraenidae (*Sphyraena ensis*), Malacanthidae (*Caulolatilus affinis*), Stromatidae (*Peprilus medius*), Ariidae (*Bagre marinus*), Gerridae (*Guerres cinereus*) and Mugilidae (*Mugil curema*); were represented by one species.

Among areas, family and species diversity showed higher numbers in the Gulf of Panama (14 families and 34 species) while 7 families and 10 species were registered in the Gulf of Chiriquí. Among landing sites, higher diversity values were found in San Felipe market with 18 species, Coquira Port with 17 species and Anton fishery market with 11 species. All these landing sites are in the Gulf of Panama. In contrast, landing sites located in the Gulf of Chiriquí showed 5 and 10 species in Port of Montijo and Port of Pedregal, respectively (Table 1). An interesting finding was the existence of species from the Atlantic side of the country in Pacific Landing sites, i.e., *Mycroperca xenarcha*, *Paralonchurus brasiliensis* and *Lobotes surinamensis*.

### 3.3. Genetic Diversity

COI haplotypes were identified for all species. Nonetheless, genetic diversity values such as nucleotide diversity (π) and haplotype diversity (*Hd*) were quantified in species with five or more individual sequences. Genetic diversity comparisons between sampling sites were only conducted in species with representative sequences in different Gulf areas (i.e., *Lutjanus guttatus* 10 sequence in the Gulf of Panama, 10 sequences in the Gulf of Chiriquí and five in the Gulf of Montijo and 11 from Coiba (Table 2). Overall genetic diversity parameters were in a range between 0.54 and 1.00 for haplotypic diversity and between 0.00113 and 0.01656 for nucleotide diversity.

#### 3.3.1. Genetic Connectivity and Demographic History

Overall values of *Fst* (genetic differentiation) by species, *Fst per pairs* (genetic differentiation between pairs of populations) and genetic distances (*Da*) were used as indirect values of genetic connectivity by species and among study areas. *Fst per pairs* estimates were only conducted on species with representative number of sequences in all areas (Gulf) (Table 3).

The demographic history of the most common species was determined using Tajima’s D and Fu’s values. Results from these parameters are presented in Table 2. Tajima’s D values were negative in five of the analyzed species. However, was only negative and significant in two snapper species: *Lutjanus peru* and *Lutjanus argentiventris*.

#### 3.3.2. Developing Tools for Molecular Taxonomy and Traceability

Five primer sets (forward and reverse) and five double-dye species-specific FAM-TAMRA probes were designed and tested in silico for five species: *Lutjanus guttatus*, *Lutjanus peru*, *Scomberomorus sierra*, *Caranx caballus*, *Caranx caninus* (Table 4). The average amplicon size for these species was 119 bp, with a range between 97 and 168 bp and a probe size that varied between 20 and 35 bp (Table 4).

## 4. Discussion

### 4.1. Species Diversity

Populational and taxonomic studies focused on the Pacific Panama fish fauna have been conducted in specific areas of Panama such as the Gulf of Montijo [41,42], Coiba National Park [43], Bahía Honda, the Gulf of Chiriquí [44] and Las Perlas Archipelago, with the Gulf of Panama [45] focusing mainly on taxonomic species diversity. Recently, the first taxonomic list of commercial species for the Pacific reported a total of 183 teleost species [10]. This work from Garces 2021 listed the four most diverse commercial fish families that are consistent with our results: Carangidae, Scianidae, Lutjanidae, and Haemulidae. Other Pacific side fish studies that have used also traditional taxonomic methods reported 28 commercial species belonging to the following families: Scianidae, Lutjanidae, Mugilidae, Scombridae and Serranidae [46]. In addition, fish community descriptions performed with multiple traditional collection techniques have been conducted for Coiba National Park, showing 27 commercial species within the families: Carangidae, Lutjanidae and Haemulidae out of 156 collected fish species including reef and non-commercial species [43].

In this study, 34 fish species were identified out of 164 sequences. Therefore, this is the first molecular identification study and largest number of DNA sequences deposited in Genebank for Panama Pacific marine commercial teleost species in a single publication including the main fishery areas of Panama Pacific. No similar data set have been reported for the Eastern Tropical Pacific. However. There is a similar study focused on Caribbean reef fishes; however, that study does not include Panama samples [46]. This research is also pioneer because we provide the first molecular list of commercial bony fishes collected and commercialized along the main fishery areas of the country including different points of the commercial chain, i.e., landing sites and fish markets. The total number of species and families obtained in this study represent a high percentage of the total commercial species already reported for Pacific side using traditional taxonomic methods.

Regarding feeding habits, most samples corresponded to carnivores and piscivores species, with coastal and demersal habits (Lutjanidae, Scombridae, Scianidae, Serranidae, Haemulidae and Centropomidae). Nonetheless, large pelagic carnivores with more oceanic and epipelagic habits were also molecularly identified, i.e., Jacks and Pompanos (Carangidae) barracudas (Sphyraenidae) and tunas (Scombridae).

Among the registered families of carnivore fish, the most diverse was Carangidae with four species: *Caranx caninus* (Pacific crevalle jack), *Caranx vinctus* (Cocinero jack), *Caranx caballus* (Green jack) and *Selene peruviana* (Pacific moonfish). For corvinas and croakers (Scianidae) five species were identified *Cynoscion othonopterus* (Gulf weakfish); *Nebris occidentalis* (Pacific smalleye croaker); *Cynoscyon phoxocephalus* (Cachema weakfish), *Cynoscion leiarchus* (Smooth weakfish) and *Paralochurus dumerilli* (Suco croaker). For snappers, four species were collected and molecularly identified. One was an exclusively estuarine species *Lutjanus argentiventris* (yellow snapper) [47]; one coastal species, *Lutjanus guttatus* (spotted rose snapper) [48] and two species with oceanic habits *Lutjanus peru* (Pacific red snapper) and *Lutjanus colorado* (colorado snapper) were also identified [49].

Other important families were Scombridae, Haemulidae and Serranidae. Among these families the most common species was *Scomberomorus sierra*. As a juvenile, *Scomberomorus sierra* is common in lower estuary areas. However, as it becomes an adult, it starts migration to open coastal and oceanic areas [42]. Haemulidae (grunts) family was represented by three species mainly located in the Gulf of Panama.

Other families with different feeding habits than midsize carnivores were collected in small numbers such as: Stromatidae, Lobotidae, Sphyraenidae, Ariidae, Mugilidae, Gerreidae, and Malacanthidae for a total of 14 families and 34 species. Ecologically, these feeding habits results show that fishing pressure is mainly centered on secondary consumers, large pelagics, demersal carnivores and planktivores as an evidence of a reduced selection pressure for herbivores and omnivores fish in Pacific Panamanian waters. Other non-traditionally commercial families or less popular among consumers were also molecularly identified. That was the case of Ariidae, Malacanthidae and Guerridae. Species composition of these families were very similar among Gulfs (No significant differences were observed *p* > 0.05). Nonetheless, the Gulf of Panama was slightly more diverse. Different factors could be generating a greater number of fish species in the Gulf of Panama landings.

Some other species listed in previous taxonomic and fishery studies for Panama Pacific were not collected in the present study, e.g., *Pomadasys macracanthus, Cynoscion squamipinnis*, *Cynoscion albus*, *Coryphaena hippurus*, *Hyporthodus acanthistius*, *Hyporthodus niphobles*, *Scomber japonicus*, *Scomberomorus maculatus*, *Sarda orientalis*, *Katsuwonus pelamis*, *Thunnus alalunga*, *Thunnus obesus* and *Thunus albacares.* One possible explanation to that is that most all of our samples were collected during visits to landing sites, markets and gathering sites with the exception of red snapper samples from Coiba National Park, collected from artisanal fishermen. The absence of large pelagic species such as tunas, dolphin fish and bonito in our analyzed landing sites and fishery markets obeyed to a direct commercialization of these species from the industrial fishery sector to international boats and external processing plants.

Another finding was the existence species reported in the Caribbean side of the country such as *Mycroperca xenarcha*, *Paralonchurus dumerilli* and *Lobotes surinamensis* in gathering sites and public markets of the Pacific side of the country as evidence for local commercialization and traceability of Caribbean species into bigger Pacific markets with a higher consumer demand.

Genetic diversity: understanding genetic diversity patterns can help us to understand sustainability of fisheries and fitness. Higher values of this parameter indicate which species are such asly to adapt to future disturbance. Genetic diversity is also a measure of reproduction strategy, fecundity, effective population size [50,51] and life history variation base on habitat type and period of larval development [52]. Genetic diversity values make it possible to establish comparisons of these parameters for species occupying different habitats or having different life histories [52,53,54]. Despite its importance, genetic diversity data are limited and are regionally oriented mainly epipelagic species, i.e., tunas and jacks [30,55].

For coastal and demersal species, studies are limited to the molecular identification of species of the genus *Cynoscion* at Panama Bay [56] as well in some important commercial species such as corvinas (*Cynoscion albus*) and snappers (*Lutjanus peru* and *Lutjanus guttatus*) from Pacific Costa Rica [57].; demersal shark species such as *Mustelus lunulatus* and *Mustelus henlei* [58] and deep-sea water species [59]. Therefore, values of genetic diversity for commercial species in the Eastern Tropical Pacific are reduced and for Panama Pacific are completely absent. Thus, collected information from this research for commercial species will be valuable for the establishment of a better *species-specific* tariff code base on a binomial nomenclature species system supported by molecular data of the country (haplotypes reported in Panama) [13].

In this study, genetic diversity parameters were in a range between 0.54 and 1.00 for haplotypic diversity and between 0.00113 and 0.01656 for nucleotide diversity. Overall species with less retention and higher migratory habits showed higher values of genetic diversity and elevated connectivity as an evidence of a bigger population size [52,53,60] (Table 2). Among the five analyzed species for genetic diversity, three species from Lutjanidae family (Snappers) with different life histories showed contrast in genetic diversity values (Table 2). Genetic studies for this family in the Pacific are mainly focused on Mexico and Baja California [61]. In previous studies, *Lutjanus peru* (red snapper) showed elevated genetic diversity in Mexican populations, suggesting elevated population size and lack of genetic structure in the analyzed populations [62]. Regionally, neither population boundaries nor genetic stocks have been established along the Eastern Tropical Pacific for *Lutjanus peru* or *Lutjanus guttatus* (rose spotted snapper) [63].

In contrast, species with reduced migration (estuarine habits) reported lower values of genetic diversity and connectivity, i.e., *Lutjanus argentriventris* among snappers in this study and *Lutjanus guttatus* in a previous study [64]. These results are the first genetic data for the species and no genetic information is available for this species in GenBank, only a few unpublished sequences have been uploaded to the database.

Additionally, the Pacific sierra, *Scomberomorus sierra*, is a species of pelagic habits common in oceanic waters when is adult [33,63]. However, as juveniles the species is common in coastal zones between 0 and 15 m and in the upper zone of estuaries migrating later to open areas as adults [42].

### 4.2. Genetic Connectivity and Demographic History

The study of population genetics, demographic history and genetic connectivity has proved to be an important tool for fishery management and understanding population structure, reproduction patterns, migration, and passive larval dispersal as well as determining selection pressures that allow scientists and decision makers to generate conservation strategies in fish exploited populations [50,65]. Connectivity patterns of three species: *L. guttatus*, *Caranx caballus* and *Scomberomorus sierra* were examined. Only *L. guttatus* showed significant differences among the three fishery areas (Gulf of Montijo, Gulf of Panama and Gulf of Chiriquí) (Table 3). However, elevated connectivity or lack of genetic differentiation among Gulf areas was evidenced among samples of the following species: *S. sierra* and *C. caballus* (Table 3) specially between the Gulf of Chiriquí and the Gulf of Panama. In contrast, other studies for this area also suggest elevated connectivity among the Gulf of *Chiriquí* and the Gulf of Montijo [58].

In terms of demographic history, despite *L. guttatus* showing negative values of Tajimas D (Table 2), these values were non-significant and close to 0, indicating a balancing selection, possibly associated with an intermediate population size, characteristic of a coastal demersal species that is under exploitation from both industrial and artisanal fisheries. In contrast, other snapper species such as *L. peru* considered an offshore species and *L. argentiventris* a coastal estuarine species exhibited negative and significant values of Tajima’s D (Table 2), indicating population expansion and sustainability of their fisheries in the Pacific Panama.

### 4.3. Developing Tools for Molecular Taxonomy and Traceability

Developing environmental DNA molecular markers (primers and probes) for molecular taxonomy and traceability.

Molecular data from Panama commercial fish species is not available for most of the listed and reported fish species. Recently, some species have been characterized in ceviche samples in an effort by conservation agencies and regional NGO’s, such as MarViva and COIBA AIP, to avoid the consumption of CITES and endangered shark and ray species in ceviche (Díaz-Ferguson et al. in preparation). Molecular information of invasive cichlid species including the Nile tilapia reported in Panamanian waters have been recently published [64]. Additionally, the genetic connectivity patterns and local movements of the largest fish of the world (*Rhincodon typus*) including transient individuals collected in Panama waters has been reported [66]. Environmental DNA (eDNA) or short-sized DNA fragments that an organism leaves behind in nonliving components of the ecosystems (i.e., water, sediments, or any other surface) has proved to be effective in detecting and tracking fish and fish products as well as other vertebrate species without the necessity of sample collection or even species observation [67,68,69]. Through this research we provide primers, probes and target amplicon regions for qPCR eDNA detection of five important commercial fish species (*Lutjanus guttatus*, *Lutjanus argentiventris*, *Caranx caballus*, *Caranx caninus*, *Scomberomorus sierra*). All designed amplicons had an average size of 119 base pairs (Table 4). These primers and probes could be used as a methodology to avoid exportation fraud and the use of false tariff codes for exportation. Detection of amplicons this size for eDNA is documented to be more efficient than bigger fragments. However, small sequences might not be fully sequenced for confirmation, and ambiguities within the sequences are common to the degraded nature of these products especially when samples are coming from processed foods such as liver, oil pills, shark fin soup, skin care products [70] and ceviche [13]. In addition, of the new developed markers, new obtained haplotypes were deposited on Genebank for all analyzed species. This information will become a reference for fishery studies providing the origin and fishing area of these species. Additionally, fish mitochondrial haplotypes have been used for determining and tracking farm fish population origins as well as to introduce new variants into hatcheries to improve genetic variability, fitness, and breeding programs [71].

## 5. Conclusions

Conservation and management of genetic diversity is key for sustainable fisheries. This research provides important information and molecular tools for accurate identification of species, its traceability along the supply chain by providing species-specific tariff codes using a binomial species-specific nomenclature system, conducting species-specific biodiversity monitoring and fishery management. In addition, this information contributes to the future generation of a molecular database for accurate stock identification, genetic health programs, the establishment of population boundaries, connectivity patterns, understanding population patterns and effective management of population size (Ne). Better understanding of these features will provide accurate information for decision makers and policy planners that contribute and support national and regional policies based on scientific data and create a balance between the power of the industrial fishery sector, local governments, and fishery managers. Thus, by providing the first haplotypes and values of genetic diversity for some species such as *L. guttatus*, *S. sierra* and *C. caballus* collected in Panamanian waters, we form the molecular database that will be a step forward for the future establishment of a marine species and fishery product traceability program that will be crucial to achieving an ecosystem management approach to our fisheries, recovering overexploited stocks and reducing illegal, unreported and unregulated fisheries (IUU) [23,72].

This work represents the first attempt at providing a molecular database (GeneBank accession numbers) of Panamanian commercial fisheries, i.e., fish species and fishery products detected molecularly. This information is the first step toward the establishment of a future fish traceability program in the country based on a BNS and supported by molecular data. These data will be key not only for the traceability of commercial species but also for future genetic stock assessment, the establishment of conservation units and sustainable fishery programs. In addition, having a strong traceability system will strengthen the Hazard Analysis and Critical Control Point (HACCP) system of the country and minimize the impact of (IUU) in the transparency of our fisheries [73]. This research also contributes with the first species-specific primers and probes designed in silico for environmental DNA (eDNA) detection and traceability of some of the most common commercial fish species reported in Panama waters. The production of molecular markers for eDNA detection of commercial species will be the base for a future molecular traceability pilot program in Panama with the potential to become a model for Latin America and the Greater Caribbean region. In the future, similar efforts should be considered for sharks, rays and billfishes since several populations are currently endangered, banned or has CITES status.

## Figures and Tables

**Figure 1 animals-13-02272-f001:**
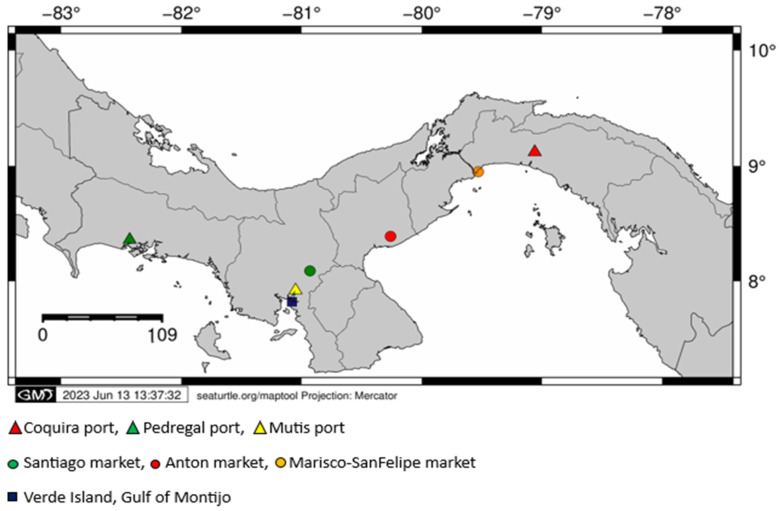
Map of the Republic of Panama, showing collection sites (ports associated to landing sites in red and gathering centers in blue) per gulf along the Pacific Panama.

**Table 1 animals-13-02272-t001:** Teleost fish families and species identified per site, common name, and GenBank Accession number.

Site	Family	Species	Common Name	GenBank Accession Number
GP _1_SFelipe	Scombridae	*Scomberomorus sierra*	Pacific Sierra	OQ790166–OQ790174
		*Euthynnus affinis*	Tuna	OQ844071
	Carangidae	*Caranx vinctus*	Cocinero	OQ790077
		*Caranx caninus*	Pacific Crevalle Jack	OQ801572
		*Centropomus unionensis*	Bigeye trevally	OQ850133
		*Caranx caballus*	Green Jack	OQ789895
		*Selene peruviana*	Peruvian moonfish	OQ843845
	Haemulidae	*Pomadasys panamensis*	Panama grunt	OQ789718
	Lobotidae	*Lobotes surinamensis **	Atlantic Triple Tail	OQ844072
		*Lobotes pacificus*	Pacific Triple Tail	OQ844073
	Serranidae	*Mycteroperca xenarcha **	Caribbean grouper	OQ844067
		*Hyporthodus acanthistius*	Pacific grouper	OQ844070
	Sphyraenidae	*Sphryraena ensis*	Pacific barracuda	OQ850073
	Scianidae	*Nebris occidentalis*	Guabina	OQ843562
		*Cynoscion phoxocephalus*	Cachema weakfish	OQ843904
	Lutjanidae	*Lutjanus guttatus*	Rose spotted snapper	OQ850300
		*Lutjanus argentiventris*	Yellow snapper	OQ790134–OQ790136
	Melacanthidae	*Caulolatilus affinis*	Bighead tilefish	OQ844074
GP_2_Anton	Scombridae	*Scomberomorus sierra*	Sierra fish	OQ790166–OQ790174
	Carangidae	*Caranx caballus*	Green Jack	OQ789895
		*Caranx caninus*	Pacific Crevalle Jack	OQ801572
		*Selene peruviana*	Peruvian Moonfish	OQ843845
	Centropomidae	*Centropomus robalito*	Yellowfin Snook	OQ844075
	Lutjanidae	*Lutjanus guttatus*	Spotted rose snapper	OQ850300
	Scianidae	*Cynoscion othonopterus*	Gulf weakfish	OQ850144
		*Ophioscion scierus*	Tuza croaker	OQ848437
		*Paralonchurus dumerilii*	Banded croacker	OQ848738
	Stromateidae	*Peprilus medius*	Pacific Harvest Fish	OQ848659
GP_3Coquira				
	Carangidae	*Caranx caninus*	Pacific crevalle Jack	OQ801572
		*Caranx caballus*	Green Jack	OQ789895
	Centropomidae	*Centropomus viridis **	Common Snook	
		*Centropomus medius*	Blackfin snook	OQ848755
	Lutjanidae	*Lutjannus gutattus*	Rose spotted snapper	OQ850300
	Haemulidae	*Haemulopsis leuciscus*	Raucous grunt	OQ849145
		*Orthopristis chalceus*	Brassy grunt	OQ849164
	Scianidae	*Nebris occidentalis*	Pacific smalleye croaker	OQ843562
		*Cynoscion phoxocephalus*	Cachema weakfish	OQ843904
		*Cynoscion leiarchus **	Smooth weak fish	
		*Larimus pacificus*	Pacific drum	OQ849155
	Guerridae	*Gerres simillimus*	Yellow fin mojarra	OQ872776
	Lobotidae	*Lobotus pacificus*	Pacific Triple Tail	OQ844073
	Ariidae	*Bagre marinus **	Gafftopsail catfish	
	Mugilidade	*Mugil curema **	White mullet	
GCH_1-Pedregal				
	Carangidae	*Caranx caballus*	Green Jack	OQ789895
	Lutjanidae	*Lutjanus argentiventris*	Yellow snapper	OQ790134–OQ790136
		*Lutjanus guttatus*	Rose spotted snapper	OQ850300
		*Lutjanus peru*	Red Snapper	OQ791284
	Scombridae	*Scomberomorus sierra*	Pacific sierra	OQ790166–OQ790174
GMon_1-Mutis				
	Scombridae	*Scomberomorus sierra*	Pacific sierra	OQ790166–OQ790174
	Scianidae	*Cynoscion phoxocephalus*	Cachema weakfish	OQ843904
	Stromatidae	*Peprilus medius*	Pacific Harvest fish	OQ848659
	Lutjanidae	*Lutjanus argentiventris*	Yellow snapper	OQ790134–OQ790136
		*Lutjanus guttatus*	Rose spotted snapper	OQ850300
GMon_2-Santiago				
	Carangidae	*Caranx vinctus*	Jack	OQ790077
		*Caranx caninus*	Pacific Crevalle Jack	OQ801572
		*Caranx caballus*	Green Jack	OQ789895
	Lutjanidae	*Lutjanus Colorado*	Colorado snapper	OQ801537–OQ801540
GCH_PixCOIBA	Lutjanidae	*Lutjanus guttatus*	Rose spotted snapper	OQ850300
		*Lutjanus peru*	Red Snapper	OQ791284

Note: GP (Gulf of Panama); GCH (Gulf of Chiriqui); GMon (Montijo area); GCH-PNC (Coiba); sequences with (*) have been edited and verified to create their accession numbers. However, accession numbers have not been assigned yet.

**Table 2 animals-13-02272-t002:** Overall values of genetic diversity and Tajima’s D of the six most common teleost fish species sampled in ports and markets of the Pacific Panama.

Common Name	Species	Familia	Total de Secuencias	Haplotype Diversity (*Hd*)	Nucleotide Diversity (*π*)	Tajima’s D
Spotted rose snapper	*Lutjanus guttatus*	Lutjanidae	36	0.775	0.020	−0.155
Yellow snapper	*Lutjanus argentiventris*	Lutjanidae	12	0.63	0.031	−2.25 **
Red snapper	*Lutjanus peru*	Lutjanidae	17	0.86	0.010	−2.17 **
Pacific Sierra	*Scomberomorus sierra*	Scombridae	16	0.90	0.0045	−1.09
Green jack	*Caranx caballus*	Caranjidae	13	0.91	0.0039	−1.26
Pacific crevalle jack	*Caranx caninus*	Caranjidae	10	0.93	0.0038	−0.86

All species presented in this table showed 10 or more sequences and were represented in more than one Gulf. Only two species showed significant values of Tajima’s D *p* < 0.001 **.

**Table 3 animals-13-02272-t003:** Pairwise *Fst* between sites values for the most common taxa (all sampled individuals by gulf). Comparisons between samples of Coiba National Park were only conducted in *L. guttatus*.

*Lutjanus guttatus*	Panama	Montijo	Chiriqui	Overall by Sp. Hudson 2000 Snn *p* < 0.0001
Gulf of Panama	0.00			
Gulf of Montijo	0.087	0.00		
Gulf of Chiriqui	0.41	0.45	0.00	*L. guttatus* **
Coiba-Pixvae	0.052	0.00	0.44	
*Caranx caballus*				
Gulf of Panama	0.00			
Gulf of Montijo	0.424	0.00		*C. caballus* (*p* > 0.05 ns)
Gulf of Chiriqui	0.500	0.00	0.00	
*Scomberomorus sierra*				
Gulf of Panama	0.00			
Gulf of Montijo	0.0694	0.00		*S. sierra* (*p* > 0.05 ns)
Gulf of Chiriqui	0.0614	−0.058	0.00	

** *p* < 0.005; ns: non significant.

**Table 4 animals-13-02272-t004:** Developing real time PCR eDNA detection markers for molecular traceability of Pacific Panama commercial fish species.

Species	Primers Sequences	Probe Name and Sequence	Amplicon Size
*L. guttatus*	L-GATCGGAGGATTCGGGAACT R-GGGTAGACTGTTCACCCAGT	5′FAM-TCCAGCACCGGCTTCTACTCCA-3′TAMRA	168 bp
*L. argentiventris*	L-TACTACTCGCCTCCTCTGGA R-TGGTTAGGTCAACAGACGCT	5′FAM-TCCTGTTCCGGCACCGGCTT-3′TAMRA	108 bp
*C. caballus*	L-TGGGACTGGCTGAACTGTTT R-CCCTGCTAGGTGAAGGGAAA	5′FAM-TGCCCACGCGGGAGCATCAGT-TAMRA	97 bp
*C. caninus*	L-GCTTCTGACTTCTCCCTCCT R-GGCATGGGCAAGATTACCAG	5′FAM-AGCCTGTTCCAGCTCCGGCT-TAMRA	116 bp
*S. sierra*	L-AGCCCTTCTTGGAGATGACC R-TCAGTTTCCAAACCCTCCGA	5′FAM-ACAATGTAATCGTTACGGCCCATGCC-TAMRA	109 bp

## Data Availability

All sequences haplotypes are available in Genbank and accession numbers are provided in Table 1.

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
