# Peer review of "Building a Teleost Fish Traceability Program Based on Genetic Data from Pacific Panama Fish Markets"

_animals, 2023, doi:10.3390/ani13142272_

Round 1
Reviewer 1 Report
The manuscript titled "Building a teleost fish traceability program based on genetic data from Pacific Panama fish markets" collected fish tissue samples from 203 adult individuals in the main ports and markets along the Pacific coast of Panama. Molecular identification using the cytochrome oxidase I gene segment confirmed the presence of 34 species from 14 families, including three Caribbean species among the analyzed samples. The study is important as it provides novel genetic information on local commercial fish species, but it requires adjustments before being eligible for publication. There are some nomenclatural errors, and the main issue lies in the structure of the text. The authors need to provide introductory texts in the Introduction section, present results in the Results section, and discuss the findings in the Discussion section. These adjustments are necessary for the manuscript to meet the publication criteria.

The English language is effective in communicating the findings.
Author Response
All suggested corrections were added to a new file submitted to this website.

Author Response
All comments and edits were responded in the new file of the manuscript that was send to the editor in this website.
